# Bird Assemblages in a Peri-Urban Landscape in Eastern India

**Ratnesh Karjee** [1] , **Himanshu Shekhar Palei** [2,*] , **Abhijit Konwar** [1] , **Anshuman Gogoi** [1] and **Rabindra Kumar Mishra** [1]

1 Department of Wildlife and Biodiversity Conservation, Maharaja Sriram Chandra Bhanjadeo University, Baripada PIN-757003, Odisha, India
2 Aranya Foundation, Plot No-625/12, Bhubaneswar PIN-751019, Odisha, India
* Correspondence: himanshu.palei@gmail.com

**Simple Summary:** Globally, biodiversity is adversely affected by urbanization. To explore the effect of urbanization on bird diversity in the peri-urban landscape, we surveyed four different habitats and three seasons in Baripada, Odisha, India, using point counts along the transects between February 2018 to January 2019. During the survey, 117 bird species with a total of 6963 individuals were found in the study area, belonging to 48 families and 98 genera, with cropland areas showing the most avian diversity. Among seasons, we observed the highest bird species richness in winter and the highest similarity of species richness in monsoon and summer. Finally, our research found that agricultural landscapes play important roles in preserving bird diversity in urban landscapes. Our study can help local governments with urban planning and habitat management while preserving local biodiversity, including birds.

**Abstract:** Urbanization plays an important role in biodiversity loss across the globe due to natural habitat loss in the form of landscape conversion and habitat fragmentation on which species depend. To study the bird diversity in the peri-urban landscape, we surveyed four habitats—residential areas, cropland, water bodies, and sal forest; three seasons—monsoon, winter, and summer in Baripada, Odisha, India. We surveyed from February 2018 to January 2019 using point counts set along line transects; 8 transects were established with a replication of 18 each. During the survey, 6963 individuals of 117 bird species belonged to 48 families and 98 genera in the study area, whereas cropland showed rich avian diversity. Based on the non-parametric multidimensional scale (NMDS) and one-way ANOVA, bird richness and abundance differed significantly among the habitats. Cropland showed higher species richness than other habitats; however, water bodies showed more abundance than others. The similarity of bird assemblage was greater between residential areas and cropland than forest and water bodies based on similarity indices. Among seasons, we observed the highest bird species richness in winter and the highest similarity of species richness in monsoon and summer. In conclusion, our study reported that agricultural and degraded landscapes like cropland play important roles in conserving bird diversity in peri-urban landscapes. Our findings highlighted and identified the problems that affect the local biodiversity (e.g., birds) in the peri-urban landscape. It can assist the local government in urban planning and habitat management without affecting the local biodiversity, including birds.

**Keywords:** bird diversity; species composition; habitat characteristics; urbanization; feeding guild

## 1. Introduction

Across the tropics, one of the most significant human impacts on natural and rural areas is urbanization [1]. Urban growth may result in habitat loss and fragmentation, isolating native species genetically or demographically and reducing biodiversity [2]. Species abundance and composition can be affected by modifying the structure and function of urban space due to the continuous growth of the urban habitat [3]. Urbanization and biodiversity showed an inverse relationship, i.e., more green space encouraged high biodiversity

and vice-versa [4]. The complexity of physical, ecological, and social elements in urban areas increases the challenge of conserving and managing biodiversity [5]. Birds have long been considered bioindicators of urbanization's effects on biodiversity among wildlife groups [6,7]. As a result of replacing natural habitats with built-up areas, urbanization has led to the extinction of birds [8,9]. There has been evidence that urbanization causes an increase in exotic species and a reduction in native species diversity [8,10]. However, urban areas are also inhabited by threatened plant and animal species [11]. Various studies have found that the size of urban green spaces contributes significantly to the richness, density, and variety of bird species in urban habitats [12,13].

Habitat destruction and human disturbances decrease avian species' diversity and force them to inhibit in urban areas [14,15]. In India, the wild lands are facing tremendous anthropogenic pressure [16], which significantly impacts the structure of the avian community [17,18], as they use seeds, plants, insects, and other vertebrates or invertebrates in their diet [19]. However, the overall community structure of birds in any landscape can be assessed by monitoring the species richness and abundance on a spatio-temporal scale [14]. Such management practices may explain the role of environmental limiting parameters and anthropogenic factors' interaction in determining the diversity and density of avifauna [20]. Habitat loss and fragmentation due to anthropogenic pressures are primary drivers of global biodiversity decline [21,22]. Forest fragmentation occurs when large, continuous forests are divided into smaller blocks by roads, clearing for agriculture, urbanization, and other human activities. Urban development for residential, commercial, and industrial properties' was undeniably the most damaging, persistent, and rapidly expanding form of anthropogenic pressure [23,24].

There is a critical relationship between bird diversity and various environmental factors. Many studies discovered the mixture of bird diversity in various habitats like urban and rural habitats, farmland, and forest habitats [25–30]. It is predicted that by the year 2050, most of the global population of birds will inhabit the urban landscape [31]. Farmland, pastureland, and urban areas are important bird habitats as they hold much wildlife outside the protected areas [28,30]. Urban habitats encourage bird populations in cities and their surroundings; however, the urbanization process, like landscape conversion, is a great threat to the bird population [15,32]. The growing human population and rapid landscape transformation for urban uses threaten biodiversity [33].

We aimed to study the effect of urbanization on bird species diversity and richness in the peri-urban landscape of Baripada, Odisha, India. The study also examined the effect of different habitats, viz. residential areas, cropland, water bodies, and forest area, and seasons on avian diversity and richness. We hypothesized that habitat heterogeneity describes significant differences in species diversity and richness. Because of the presence of more microhabitats and away from the urban center, we predicted that the variety of bird assemblages might be higher in cropland habitats than in other habitats.

## 2. Materials and Methods

### 2.1. Study Area

This study was carried out in Baripada, which lies between 21.90°–21.96° N and 86.71°–86.78° E. It is located in the Chotanagpur Plateau Region of Odisha, Eastern India, with an altitude of 45 m a.s.l (Figure 1). It has a tropical climatic condition that experiences an extremely hot and humid summer (45 °C) followed by a humid monsoon (30 °C) and chilling winter (10 °C) with an annual temperature of 30 °C. The winter season is observed between November to February. After that, summer continues from March to June, followed by the monsoon season from July to October, with an annual mean rainfall of 1800 mm. The city area is spread over approximately 30 km², with a population of 116,874 (Census of India 2011). The study area consists of diverse habitats, such as highly urbanized residential and commercial complexes, agricultural lands, and woodlands. The dominant vegetation in urban and suburban areas include *Ficus benghalensis*, *Ficus religiosa*, *Magnifera indica*, *Azadirachta indica*, *Aegle marmelos*, *Tamarindus indica*, etc. In forest areas, the

vegetation is dominated by *Shorea robusta*, *Terminalia bellirica*, *Cassia fistula*, *Suzygium cumini*, *Bombax ceiba*, etc. Invasive shrubs such as *Chromolaena odorata* and *Lantana camara* are spread in natural vegetation patches as well as urban green spaces. The agricultural lands present in this area are used only once a year to cultivate paddy, i.e., from July to December, and the rest of the year, it remains abandoned. Our study focused on Baripada and its outskirts surrounded by woodlands and agricultural lands with human interference [34].

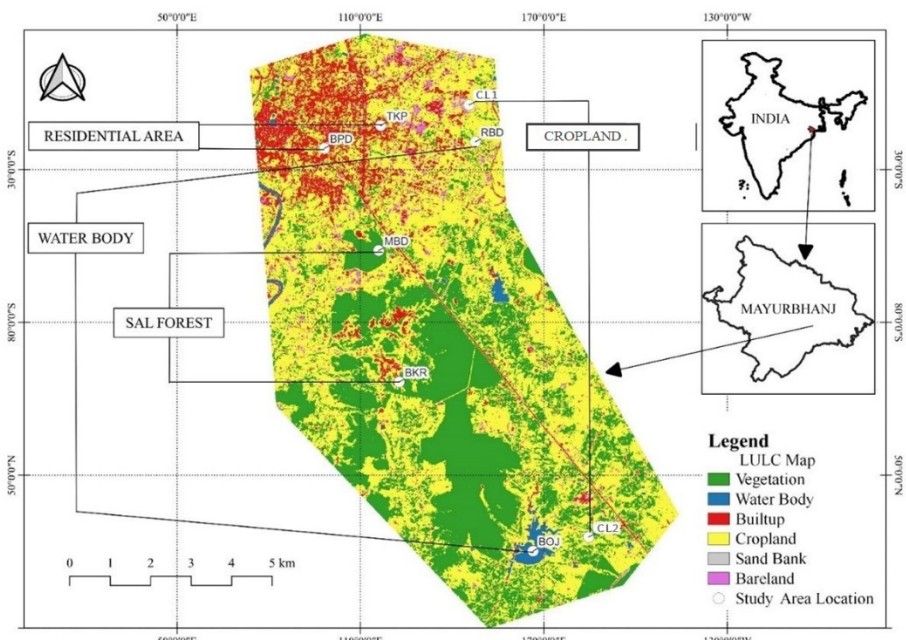

**Figure 1.** Map showing the different habitats and transect locations in the city of Baripada, Odisha, India—(a) transect 1 at Takatpur (TKP), (b) transect 2 at Baripada (BPD), (c) transect 3 at Rani Bandh (RBD), (d) transect 4 at Borjhor (BOJ), (e) transect 5 at cropland (CL1), (f) transect 6 at cropland (CL2), (g) transect 7 at Manchabandha (MBD), and (h) transect at Budhikhamari Forest (BKR). Refer Figure 2 for more information.

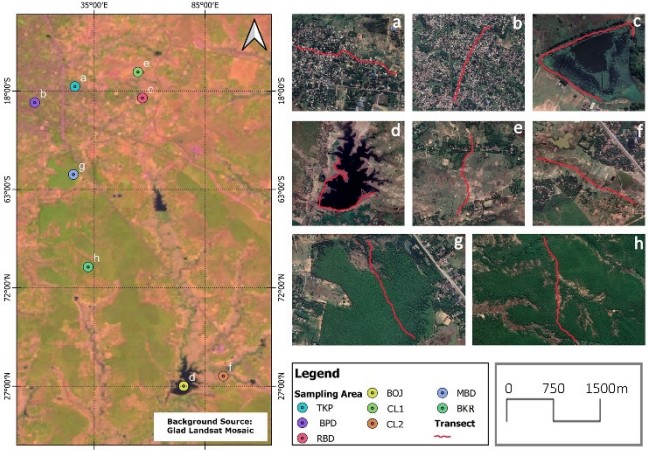

**Figure 2.** Map showing all eight transects in the study area; red color represents the transect line; (**a**) TKP = Takatpur, (**b**) BPD = Baripada, (**c**) RBD = Rani Bandh, (**d**) BOJ = Borjhor, (**e**) CL1 = Cropland1, (**f**) CL2 = Cropland2, (**g**) MBD = Manchabandha, (**h**) BKR = Budhikhamari.

We selected four habitat types for bird sampling in a peri-urban landscape of Baripada, two transects in residential areas (RA), two in water bodies (WB), two in Sal (*Shorea robusta*) forest habitats (SF), and two in cropland habitats (CL) (Figures 1 and 2, Table 1).

**Table 1.** Description of different habitat types in the study area.

| Habitat | Transect | GPS Coordinate | | No. of Points | Transect Length (km) | Characteristics of the Study Area |
|---|---|---|---|---|---|---|
| | | Lat (N) | Long (E) | | | |
| RA | BPD | 21.928153 | 86.737023 | 3 | 0.6 | Human-dominated landscapes with fewer vegetation covers like *Shorea robusta*, *Mangifera indica*, *Ficus benghalensis*, etc. |
| | TKP | 21.933584 | 86.750474 | 3 | 0.6 | Human-dominated landscape with vegetation covers like *Bombax ceiba*, *Bamboosa* sp., *Gmelina arborea*, *Mangifera indica*, *Lantena camara*, etc. |
| WB | RBD | 21.929695 | 86.773065 | 2 | 0.4 | A small pond with aquatic flora and medium patches of vegetation covers like *Borassus flabellifer*, *Lantena camara*, *F. bengalensis*, like *Nimphea* sp., etc. |
| | BOJ | 21.831909 | 86.78687 | 6 | 1.1 | Large ponds with aquatic flora and small vegetation covers like *Nimphea* spp., *Hydrilla* sp., *Utricularia* sp. *Ipomea* sp., etc. |
| CL | CL1 | 21.938516 | 86.771574 | 3 | 0.6 | Mosaic landscape of annual crops (paddy) or fallow land with small patches of vegetation, grass, seasonal canals, and ditches. |
| | CL2 | 21.83528 | 86.800143 | 2 | 0.5 | Mosaic landscape of annual crops or fallow land with small patches of vegetation, grass, seasonal ditches |
| SF | MBD | 21.903739 | 86.749982 | 3 | 0.6 | Sal-dominated forest covers with small canals and open areas with scattered patches of *Ziziphus jujuba* |
| | BKR | 21.872335 | 86.754893 | 3 | 0.6 | Sal-dominated forest covered with small canals and open areas with *Suzygium cumini* and *Ziziphus jujuba*. |

RA = Residential Areas, WB = Waterbodies, CL = Cropland, SF = Sal Forest, BPD = Baripada, TKP = Takatpur, RBD = Rani Bandh, BOJ = Borjhor Dam, CL1 = Cropland near University, CL2 = Cropland near Borjhor, MBD = Manchabandha Forest, BKR = Bhudikhamari Forest.

### 2.2. Field Survey and Avifaunal Sampling

We studied bird diversity and abundance from February 2018 to January 2019 in the study area using point counts set along line transects [35], following an approach comparable to that used in other large-scale bird surveys [36]. A total of 8 transects (2 transects in each habitat) were established, with 18 replications in each transect to obtain a spatially homogenous distribution (Figures 1 and 2, Table 1). The total length of all transects was 5 km (0.62 mean; SD ± 0.20 km, 0.40–1.1 km range) long (Figure 2, Table 1). They were demarcated before surveys using 1:25,000 topo maps, aiming to obtain a sufficient length route as linear and continuous as possible. Within each transect at each habitat, we established permanent sampling points (Number of sampling points depending upon the length of the transect) for the bird count. In each transect, the first sampling point coincided with the beginning of the transect; all the other points were set 200 m apart. This spacing was considered sufficient to avoid double counts [37]. In all, 25 sampling points were established in the study area, with 6 being in residential areas, 8 near water bodies, 5 in cropland, and 6 in sal forest habitat. In the water body habitat, the line transects, and sampling points were established at the edge of the reservoir. Waterbird ground counts are conducted in accordance with generally accepted standards in the field [3]. Bird species detectability may differ between species and habitats; therefore, we recorded birds within a circle of a 50 m radius around the observer for a specific period (10 min) at each sampling point [38–40]. We did not count birds observed between sampling points. Birds earlier recorded in the sampling points were not included in other sampling points of the transect. Overflying birds were not included, as they could only be moving through or above the surveyed habitat. Birds were counted only if they showed evidence of using the habitat. The counting of avian species was conducted during the bird's peak activity (up to 3 h

after sunrise) and in the early morning after the sun rises [38,39]. Two observers with similar training levels walked along the transects and recorded birds in sampling points. Observations of birds were not made in adverse weather conditions. Every month we established two counts in each transect at each site except for the month of January, May, June, July, August, and November (only a single count for each site and transect).

### 2.3. Nonparametric Richness Estimation

Bird diversity was assessed in terms of species richness and total abundance, considering the total number of species observed per site. We plotted a species accumulation curve to evaluate whether the number of bird species sampled was representative of the bird community. Individual-based rarefaction curves were used to compare species richness at the habitat and season levels.

### 2.4. Species Richness, Diversity, and Abundance

Species richness is estimated as the number of bird species present in a particular habitat and season.

Shannon's diversity index ($H'$) was calculated by multiplying the proportion of each species by their natural log as:

$$H' = \Sigma pi \log(ln) pi$$

Here $pi$ is the proportion ($n/N$) of individuals of a particular species found ($n$) divided by the total number of individuals recorded ($N$), $ln$ is the natural log, and $\Sigma$ is the sum of the calculations.

Similarly, to understand the dominant species within the community, the Simpson diversity index ($D$) was calculated by using the formula:

$$D = 1 - \Sigma n\,(n-1)/N\,(N-1)$$

Here $n$ is the total number of birds of a particular species and $N$ is the total number of birds of all species.

The evenness of bird species compares the similarity of the population size of each species. Evenness Index ($J'$) was calculated using the ratio of observed diversity to maximum diversity using the equation [41].

$$J' = H'/H_{max}$$

Here, $H'$ is the Shannon–Wiener Diversity index, and $H_{max}$ is the natural log of the total number of species. Rank abundance plots were constructed to investigate species abundance distributions between habitats.

Bird abundance was obtained by ranking the species according to their frequencies, and then the proportions of each species were obtained using the equation:

$$\text{Species abundance} = S_n/N$$

Here, $S_n$ = number of the bird in the reference species and $N$ = Total number of birds.

### 2.5. Bird Assemblage and Similarity

Four similarity indices were calculated to estimate shared species richness between habitats and different seasons. These included qualitative similarity estimates using the Euclidian distance, Jaccard index, and Morisita–Horn index [41,42].

Moreover, bird richness and diversity similarity were determined using Bray–Curtis similarity or distance index, which is formulated as

$$BC_{ij} = 1 - (2C_{ij}/S_i + S_j)$$

Here $i$ and $j$ are the two habitats, $S_i$ and $S_j$ are the total numbers of birds counted on $i$th and $j$th and $C_{ij}$ is the only lesser count for each bird species counted in both habitats. In the Bray–Curtis similarity index, a value nearer to 0 means the communities have the same species composition, and a value closer to 1 means no share of any species.

To examine changes in bird functional diversity among different habitats, we classified birds into various guilds based on their diets: carnivorous (C), frugivorous (F), granivorous (G), omnivorous (O), insectivorous (I), and nectarivorous (N). A heat map has been produced to understand the spatio-temporal assemblage of birds' feeding guild. Although Indian birds have mixed food habits, a simplified food guild based on the predominant food habits of each bird species was followed in this study. The classification of feeding guilds is followed from previous studies [43,44].

*2.6. Statistical Analysis*

The non-parametric multidimensional scale (NMDS) test was performed to check the significant variation of bird community among the habitats and seasons using the permutation test (999 permutations). After that, a one-way ANOVA test was run to check the variation of birds' richness and abundance in the study area. Before the ANOVA test, data normality was checked by the Shapiro–Wilk normality test. Again, a multiple comparison Tukey's test was performed to quantify variation among the habitats and seasons. The statistical analyses were performed in R 4.0.2 statistical data processing packages [45]. Species accumulation curves, diversity indices, rank abundance plots, habitat share Venn diagram, and heatmap were calculated using "BiodiversityR" [46], "VennDiagram" [47], and "superheat" [48] packages.

## 3. Results

*3.1. Species Richness and Diversity*

During the survey, 6963 individuals of 117 bird species were recorded belonging to 48 families and 98 genera within the four different habitats (Appendix A). Of these, 85.83% are the resident species (103 bird species), while 9.17% (11 bird species) are winter migrants and the rest are summer visitors (three bird species). The highest bird richness was observed in CL (64 species; evenness $J' = 0.91$), followed by RA (56 species; evenness $J' = 0.90$) and SF (54 species; evenness $J' = 0.88$) (Figure 2). The WB represents the lowest species richness (37 species; evenness $J' = 0.78$). The individual-based rarefied richness curve of bird species reached an asymptote for all habitats, indicating that the sampling effort was sufficient (Figure 3). The maximum value of the Shannon–Wiener Index was recorded in CL ($H' = 3.79$), followed by RA ($H' = 3.63$), SF ($H' = 3.51$), and lowest in WB ($H' = 2.84$). The value of Simpson's index of CL scored highest ($D = 0.97$), followed by RA ($D = 0.96$), SF ($D = 0.95$), and WB ($D = 0.90$).

Seasonally, winter harbored the highest richness (111 bird species; J = 0.85), followed by summer (106 bird species; J = 0.88) and monsoon (94 bird species; J = 0.88). However, summer exhibited maximum diversity ($H' = 4.10$), followed by winter ($H' = 4.02$) and monsoon ($H' = 4.02$). Moreover, we found that summer and winter shared equal values of the Simpson index ($D = 0.97$), while winter showed lower values ($D = 0.96$).

The NMDS results (non-metric fit, $R^2 = 0.936$, linear fit, $R^2 = 0.695$, stress = 0.252) showed that bird communities were significantly varied among habitats (MAST, F = 17.50, DF = 3, $R^2 = 0.272$, $p = 0.001$) (Figure 4). In addition, one-way ANOVA suggested that there was a significant variation in bird richness (F = 14.23, DF = 3, $p < 0.001$; F = 9.26, DF = 2, $p < 0.001$) and abundance (F = 6.58, DF = 3, $p < 0.001$; F = 16, DF = 2, $p < 0.001$) among habitats and seasons, respectively (Figure 5a,b and Table 2). Similarly, Tukey's HSD test for multiple comparisons also showed significant variation in bird richness and abundance among habitat and season, which are available in Table 2.

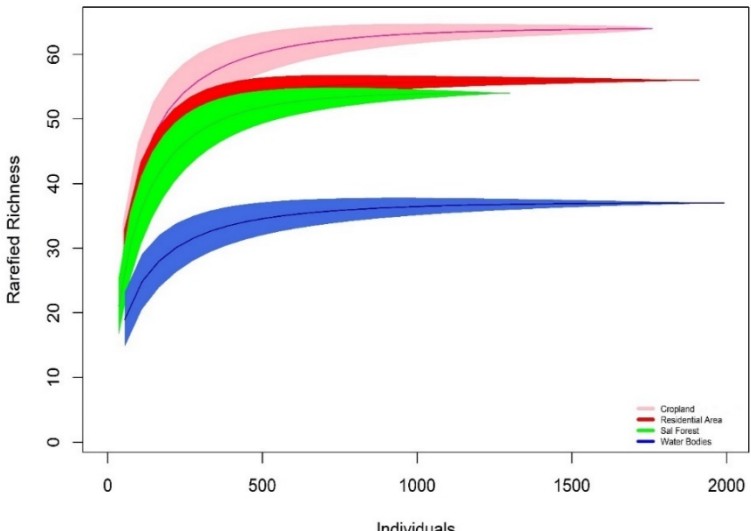

**Figure 3.** Individual-based rarefaction curve for bird species richness found in four different habitats in the study area. The shaded area around the curve indicates 95% confidence intervals (CI).

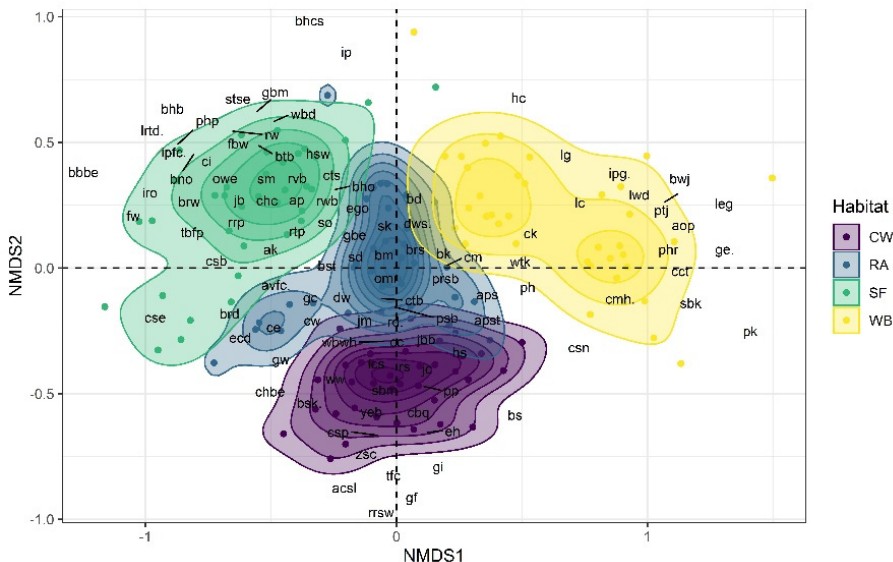

**Figure 4.** Non-metric Multidimensional Scaling (NMDS) showing the different bird species composition at each habitat (stress = 0.25; Non-metric fit $R^2$ = 0.936; linear fit $R^2$ = 0.695); CL = Cropland, RA = Residential areas, SF = Sal Forest, and WB = Water Body. Where; ap = Alexandrine Parakeet; acsl = Ashy-crowned Sparrow Lark; asp = Ashy Prinia; ak = Asian Koel; aob = Asian Openbill; aps = Asian Palm Swift; ipfc = Indian Paradise-flycatcher; apst = Asian Pied Starling; bm = Bank Myna; bs = Barn Swallow; bw = Baya Weaver; bbs = Bay-backed Shrike; bd = Black Drongo; bhm = Black-headed Munia; bk = Black kite; bwk = Black-winged Kite; bhcs = Black-headed Cuckooshrike; bho = Black-hooded Oriole; bno = Black-naped Oriole; brw = Black-rumped Woodpecker; rp = Rock Pigeon; btb = Blue-throated Barbet; bbbe = Blue-bearded Bee-eater; bq = Common Buttonquail; bst = Brahminy Starling; brd = Bronzed Drongo; bwj = Bronze-winged Jacana; brs = Brown Shrike; bhb = Brown-headed Barbet; ce = Cattle Egret; chbe = Chestnut-headed Bee-eater; cts = Chestnut-tailed Starling; cc = Common Chiffchaff; cct = Common Coot; chc = Common Hawk Cuckoo; ch = Common Hoopoe; ci = Common Iora; ck = Common Kingfisher; cmh = Common Moorhen; cm = Common Myna; csp = Common Sandpiper; csn = Common Snipe; ctb = Common Tailorbird; csb = Coppersmith Barbet; ipg = Indian Pygmy-goose; cse = Crested Serpent Eagle; cw = Citrine Wagtail, *Motacilla citreola*; dw = Dusky Warbler; rcd = Red Collared Dove; ecd = Eurasian Collared Dove; ego = Eurasian Golden Oriole; fw = Forest Wagtail; fbw = Fulvous-breasted Woodpecker;

gc = Greater Coucal; ggbw = Greater Golden-backed Woodpecker; gbe = Green Bee-eater; gbm = Green-billed Malkoha; gw = Grey Wagtail; hc = House Crow; hs = House Sparrow; hsw = House Swift; inj = Indian Nightjar; ph = Indian Pond Heron; iro = Indian Robin; irl = Indian Roller; jc = Jacobin Cuckoo; jb = Jerdon's Baza; jbb = Jungle Babbler; jm = Jungle Myna; jcs = Large Cuckooshrike; ge = Great Egret; lwd = Lesser Whistling Duck; leg = Little Egret; lg = Little Grebe; omr = Oriental Magpie-Robin; osl = Oriental Sky Lark; owe = Oriental White-eye; pfp = Paddy-field Pipit; ptj = Pheasant-tailed Jacana; pk = Pied Kingfisher; pp = Plain Prinia; php = Plum-headed Parakeet; psb = Purple Sunbird; prsb = Purple-rumped Sunbird; lrtd = Greater Racket-tailed Drongo; rrsw = Red-rumped Swallow; rvb = Red-vented Bulbul; rwb = Red-whiskered Bulbul; rrp = Rose-ringed Parakeet; rs = Rosy Starling; rtp = Rufous Treepie; rw = Rufous Wood-pecker; sbm = Scaly-breasted Munia; sk = Shikra; stse = Short-toed Snake Eagle; sd = Spotted Dove; so = Spotted Owlet; sbk = Stork-billed Kingfisher; tbfp = Thick-billed Flowerpecker; tfc = Taiga Flycatcher; avfc = Asian Verditer Flycatcher, *Eumyias thalassinus*; wbwh = White-breasted Waterhen; wtk = White-throated Kingfisher; ww = White Wagtail; wbd = White-bellied Drongo; wds = Dusky Woodswallow; yeb = Yellow-eyed Babbler; yfgp = Yellow-legged Green Pigeon; zsc = Zitting Cisticola; rwl = Red-wattled Lapwing; lc = Little Cormorant; phr = Purple Heron; jlb = Jerdon's Leafbird; sm = Scarlet Minivet, *Pericrocotus speciosus*; gf = Grey Francolin; ip = Indian Pitta; gi = Glossy Ibis.

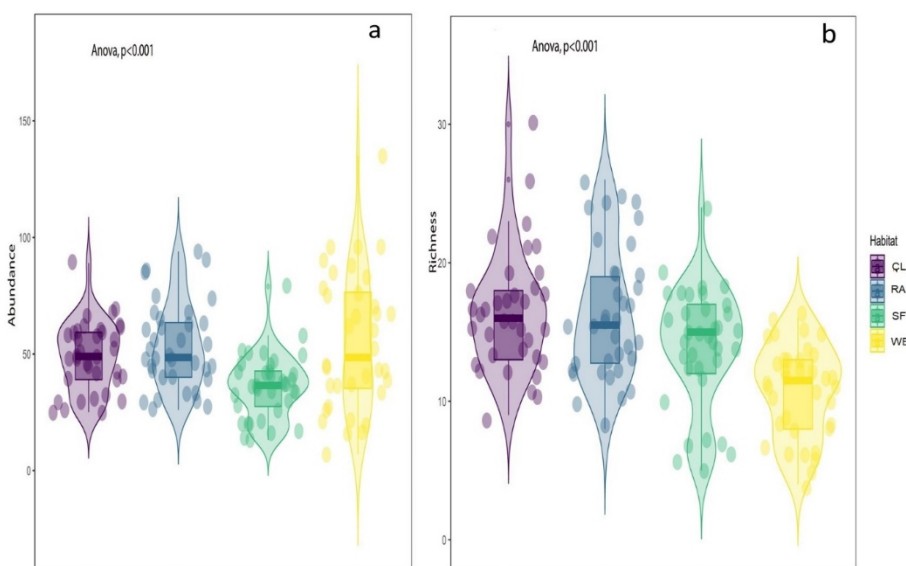

**Figure 5.** ANOVA showing that there was a significant difference between habitat in terms of bird abundance (**a**) and richness (**b**); CL = Cropland, RA = Residential areas, SF = Sal Forest, and WB = Water Body. Check Table 2 for more details.

**Table 2.** Table showing the results—(a) One-way ANOVA test for both abundance and richness. Additionally, pairwise comparisons—(b) Tukey's HSD test for bird richness and abundance across the habitat and season are represented below the ANOVA results; CL = Cropland, RA = Residential areas, SF = Sal Forest, and WB = Water Body.

| | df | Sum Sq | Mean Sq | F value | Pr (>F) | Significance |
|---|---|---|---|---|---|---|
| **ANOVA for Abundance (Habitat)** | | | | | | |
| Habitat | 3 | 7999 | 2666.4 | 6.582 | 0.000341 | <0.001 |
| Residuals | 140 | 58,714 | 405.1 | | | |
| **ANOVA for Richness (Habitat)** | | | | | | |
| Habitat | 3 | 764.7 | 254.92 | 14.23 | $3.78 \times 10^{-8}$ | <0.001 |
| Residuals | 140 | 2707 | 17.91 | | | |
| **ANOVA for Abundance (Season)** | | | | | | |
| Season | 2 | 12,048 | 6024 | 16 | $5.4 \times 10^{-7}$ | <0.001 |
| Residuals | 141 | 53,078 | 376 | | | |

**Table 2.** *Cont.*

| ANOVA for Abundance (Habitat) | | | | | | |
|---|---|---|---|---|---|---|
| ANOVA for Richness (Season) | | | | | | |
| Season | 2 | 380.4 | 190.22 | 9.26 | 0.00016 | <0.001 |
| Residuals | 141 | 2895.7 | 20.54 | | | |
| Group | 95% Confidence level of interval for mean abundance | | | | Significance | |
| | Differences | Upper | Lower | *p* Adj | | |
| RA-CL | 4.22 | −8.11 | 16.55 | 0.81 | No | |
| SF-CL | −12.80 | −25.14 | −0.47 | 0.03 | <0.05 | |
| WB-CL | 6.44 | −5.89 | 18.77 | 0.52 | No | |
| SF-RA | −17.02 | −29.36 | −4.692 | 0.002 | <0.05 | |
| WB-RA | 2.22 | −10.11 | 14.55 | 0.96 | No | |
| WB-SF | 19.25 | 6.914 | 31.58 | 0.0004 | <0.001 | |
| | 95% Confidence level of interval for mean richness | | | | | |
| RA-CL | −0.11 | −2.70 | 2.48 | 0.99 | No | |
| SF-CL | −2.5 | −5.09 | 0.09 | 0.06 | No | |
| WB-CL | −5.66 | −8.26 | −3.07 | 0.01 | <0.001 | |
| SF-RA | −2.38 | −4.98 | 0.20 | 0.08 | No | |
| WB-RA | −5.55 | −8.14 | −2.96 | 0.01 | <0.001 | |
| WB-SF | −3.16 | −5.76 | −0.57 | 0.01 | <0.05 | |
| Seasons | 95% Confidence level of interval for mean abundance | | | | | |
| Summer-Monsoon | 0.62 | −8.75 | 10.00 | 0.98 | No | |
| Winter-Monsoon | 19.70 | 10.32 | 29.08 | 0.001 | Yes | |
| Winter-Summer | 19.08 | 9.70 | 28.46 | 0.001 | Yes | |
| | 95% Confidence level of interval for mean richness | | | | | |
| Summer-Monsoon | 1.14 | −1.04 | 3.33 | 0.43 | No | |
| Winter-Monsoon | 3.87 | 1.68 | 6.06 | 0.001 | Yes | |
| Winter-Summer | 2.72 | 0.53 | 4.92 | 0.01 | Yes | |

### 3.2. Bird Rank-Abundance

A total of 6963 individual birds were counted, including 1992 in WB (28.61%), 1912 in RA (27.46%), 1760 in CL (25.28%), and 1299 in SF (18.66%) (Figure 6). Lesser Whistling Duck (*Dendrocygna javanica*), Rock Dove (*Columba livia*), Cattle Egret (*Bubulcus ibis*), and Chestnut-tailed Starling (*Sturnia malabarica*) were the dominating species for WB, RA, CL, and SF, respectively (Figure 6). Bird abundance was higher in winter, with 2957 individuals, followed by summer (2026 individuals) and monsoon (1980 individuals). Moreover, Lesser-whistling Duck was the most dominant individual in winter (364 individuals), followed by Cattle Egret in both summer (127 individuals) and monsoon (118 individuals).

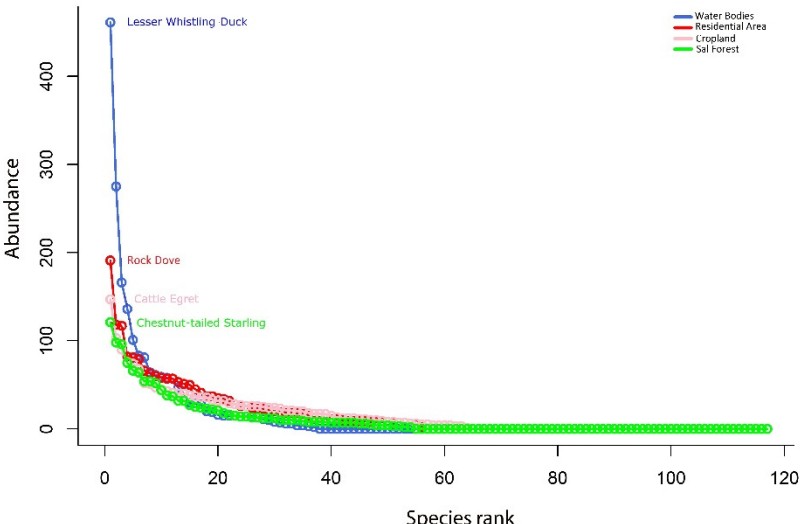

**Figure 6.** The rank-abundance represents the position of the bird in various habitats; the highest abundance of each bird in each habitat held the top position.

### 3.3. Similarity and Shared Species Richness

In our study, we recorded nine bird species Asian Pied Starling (*Gracupica contra*), Black Drongo (*Dicrurus macrocercus*), Common Myna (*Acridotheres tristis*), Citrine Wagtail (*Motacilla citreola*), Green Bee-eater (*Merops orientalis*), Red-vented Bulbul (*Pycnonotus cafer*), Red-whiskered Bulbul (*Pycnonotus jocosus*), Shikra (*Accipiter badius*), and Spotted Dove (*Spilopelia chinensis*) which are found in all habitats. The highest bird richness was shared between RA and CL, with a total of 39 species, followed by CL and SF, with 25 species. SF and WB shared very low bird species (S = 13) (Figure 7). Bray–Curtis (BC) and Jaccard Indices (JI) suggested that there were higher dissimilarities between WB and SF (BC = 0.87) and higher similarities between RA and CL (JI = 0.42). Morisita–Horn Index (MH) also revealed that the compositional similarity of birds is higher between RA and CL (MH = 0.73), followed by RA and SF (MH = 0.57), and SF and CL (MH = 0.52). The Euclidian distance index (EU) also suggested a higher similarity between RA and CL (EU = 256), followed by CL and SF (EU = 258). In contrast, RA and WB exhibited higher dissimilarity (EU = 687), followed by WB and SF (Table 3). Among seasons, the highest similarity of species richness was observed in monsoon and summer (EU = 121.30, BC = 0.18, MH = 0.94, JI = 0.87), followed by winter and summer (EU = 362.92, BC = 0.27, MH = 0.75, JI = 0.86), and winter and monsoon (EU = 376.30, BC = 0.29, MH = 0.72, JI = 0.83).

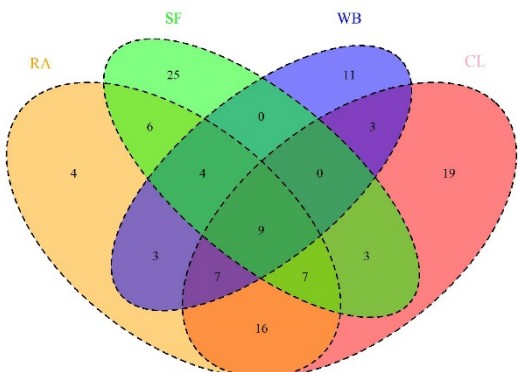

**Figure 7.** Venn diagram showing the number of unique and shared species among the different sampling habitats; CL: Cropland, RA: Residential Areas, SF: Sal Forest, WB: Waterbodies.

**Table 3.** Pairwise comparisons of the bird communities among four different habitats and seasons in the study area.

| Similarity Index | Euclidian Distance | | | Bray–Curtis | | | Morista–Horn | | | Jaccard | | |
|---|---|---|---|---|---|---|---|---|---|---|---|---|
| Habitat | RA | WB | CL | RA | WB | CL | RA | WB | CL | RA | WB | CL |
| WB | 687.18 | | | 0.81 | | | 0.19 | | | 0.33 | | |
| CL | 255.83 | 653.68 | | 0.47 | 0.78 | | 0.73 | 0.11 | | 0.48 | 0.23 | |
| SF | 305.01 | 656.31 | 281.30 | 0.54 | 0.87 | 0.66 | 0.57 | 0.07 | 0.52 | 0.31 | 0.17 | 0.19 |
| Season | Winter | Summer | | Winter | Summer | | Winter | Summer | | Winter | Summer | |
| Summer | 362.92 | | | 0.27 | | | 0.75 | | | 0.86 | | |
| Monsoon | 376.30 | 121.30 | | 0.29 | 0.18 | | 0.72 | 0.94 | | 0.83 | 0.87 | |

CL: Cropland, RA: Residential Areas, SF: Sal Forest, WB: Waterbodies.

### 3.4. Feeding Guilds and Functional Diversity

Out of a total of six feeding guilds of birds observed, the richness of insectivores (49 species; 41.9%) dominates the others, followed by omnivorous (27 species; 23.1%), carnivorous (22 bird species; 18.8%), granivorous (10 bird species; 8.5%) and frugivorous (7 bird species; 6%) (Figure 8). Only two species of nectarivorous feeding birds (Purple Sunbird *Cinnyris asiaticus*; Purple-rumped Sunbird, *Leptocoma zeylonica*) were observed. However, the individual counting of each feeding guild shows that O represents the highest

numbers (38.82%), followed by I (27.69%), C (13.54%), G (12.80%), F (4.09%), and N (3.06%), respectively. A hierarchical dendrogram cluster exhibited similarity between the guild and guild composition in our study area (Figure 9).

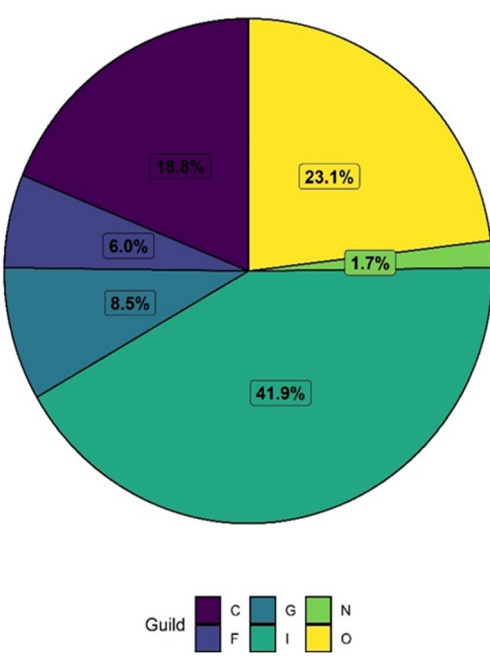

**Figure 8.** The pie chart represents the occurrence of birds feeding guild richness (%) in the study area; C = Carnivore, F = Frugivore, G = Granivore, I = Insectvore, N = Nectarivore, and O = Omnivore.

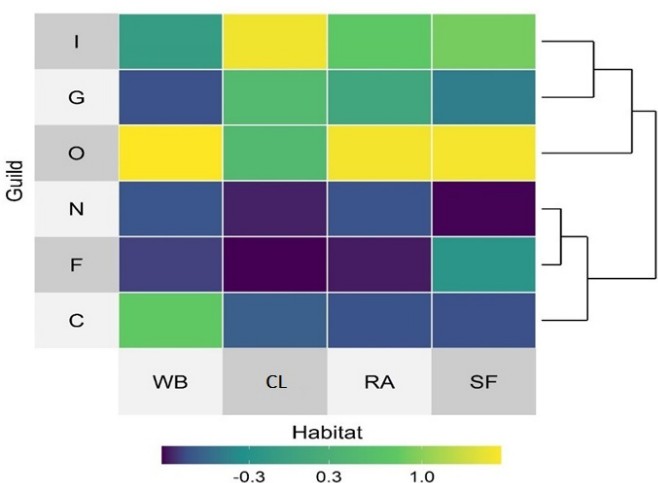

**Figure 9.** Heat map showing the distribution pattern of various feeding guilds in different habitats; blue color represented low value and yellow indicated higher value. The distribution pattern of bird guilds in various habitats formed two hierarchical clusters; CL: Cropland, RA: Residential Areas, SF: Sal Forest, WB: Waterbodies, I: Insectivore, G: Granivore, O: Omnivore, N: Nectarivore, and C: Carnivore.

## 4. Discussion

Our findings confirm our prediction that cropland supports the high diversity and richness of bird species. Worldwide, it is accepted that certain bird species favor cropland habitats considering that about one-third of the world's bird population occasionally stays in such a habitat [49,50]. For example, good management of crop fields and mosaic of micro-landscape provides food resources to promote higher bird richness [25] and agricultural habitat harboring a significant proportion of the birds' community [51]. Studies found that cropland is used as a stopover habitat by migratory bird species [52,53]. A study

from Poland reported that bird density and richness depend on the heterogeneity of the agricultural landscape [54]. In addition, some non-crops patches between the cropland support large bird numbers compared to land use diversity [55]. Furthermore, a report from Southern China suggested that bird richness significantly increased when herbaceous vegetation covers grew primarily in farmland [56]. However, in some cases, intensified agricultural production may affect farmland biodiversity. For example, after receiving new membership in the European Union, farmland bird diversity declined due to intensified agricultural activities in some European countries [57].

We found that bird richness and abundance significantly varied across the season. The change in richness and abundance of birds depends on the presence of abundant species that visit the area in different seasons [58]. Several studies argued that variations in rainfall and temperature affect the availability of food resources for birds and influence bird populations [58–60]. Sometimes, the numbers of migratory birds are maximum compared to resident birds during the migratory season affecting the bird richness and abundance [61,62]. In our study, we observed that the bird abundance was higher in winter, probably, due to the large gathering of numerous local migratory birds (e.g., Lesser-whistling Duck). Additionally, omnivorous birds were the most abundant guild compared to other guilds across the season due to their flexibility in utilizing natural resources and food [63].

Avian species shared the highest similarities among cropland and residential areas. Many bird species could be found supporting different feeding guilds in these two habitats because of the abundant food sources, accompanied by several insect prey species attracted to the crops [64–66]. Residential areas shared the similarity in abundance with cropland, which implies the component of generalist birds in the study area. The mosaic of a variety of tree covers, small water bodies, bamboo grooves, and grasses around cropland areas makes a more productive heterogeneous landscape that allows it to sustain various bird species [67,68]. However, a comparison between residential areas and forest areas showed that the diversity and richness are higher in residential areas, possibly due to habitat loss in forest areas which forced the bird species to inhibit in residential areas [14]. It is likely that well-vegetated urbanization provides bird species with refuge, nesting sites, and food sources. Studies have shown that species numbers decline with urbanization, and highly abundant species dominate the remaining species group [69–72]. In monoculture, sal-dominated forest habitats may have low fruit and flowering trees, while residential areas have parks and roadside fruit trees like banyan (*F. benghalensis*), peepal (*F. religiosa*), and jamun (*Syzygium cumini*), etc., in residential areas. Birds' diversity increased with the increase in natural woodlands; however, their diversity decreased with an increase in commercial monoculture forests [56]. In our case, we also noticed that bird diversity in sal forest was less than in residential and cropland. Probably, it is due to its dominant monoculture habitat. Likewise, small and isolated forest fragments in urban areas fail to sustain a greater diversity of birds [73].

Water bodies allowed the highest number of birds, and their abundance revealed that water bodies significantly differed from other habitats in our study area. The abundance of birds was higher in water bodies because of the large number of gathering colonial water birds like Lesser Whistling Duck (*Dendrocygna javanica*), Indian Pygmy Goose (*Nettapus coromandelianus*), Asian Openbill (*Anastomus oscitans*), etc. Birds are abundant in this habitat because of some environmental characteristics of wetlands or water bodies, like size, depth, water level, and plant species [74]. A study from Malaysia revealed that water-bird diversity, distribution, and abundance are greatly influenced by the composition and structure of vegetation and microclimatic variables. In addition, it explained that birds are adapted to a distinct set of microhabitats and microclimatic conditions [75].

The bird richness of the residential area does not exhibit any significant differences from cropland. However, total bird counts (abundance) in this habitat significantly differ from sal forest. Certain kinds of birds (e.g., House Crow *Corvus splendens*, Common Myna *Acridotheres tristis*, House sparrow *Passer domesticus*) were encouraged by urban resources. Møller [76] reported several characteristics of bird species that have adapted

to urban habitats, including large breeding ranges, a high tendency to disperse, a high rate of feeding innovation (new ways of acquiring food), and a short breeding cycle. In addition, they have a high adult survival rate and a short flight distance when approached by humans. They are likely to be more exploitative and aggressive and can adjust to this urban environment by taking advantage of the anthropological resources [77–79]. A similar pattern is found in other cities in different countries [70–72,80–83].

Our study observed that the insectivorous are the richest feeding guild, followed by omnivorous, carnivorous, granivorous, frugivorous, and nectarivorous. Insectivorous species richness has been found to coincide with food availability in agricultural and residential habitats [33,63,84]. Several studies have indicated that some groups of arthropods are more abundant in cropland and urban areas, including generalist ground arthropods, plant-feeding arthropods, and generalist pollinating and jumping spiders, so it may be this factor that has led to a predominance of insectivorous species [85–89]. However, the insectivore group did not show higher abundance in terms of the guild, which indicating reduced in their numbers, especially in residential areas, probably due to anthropogenic disturbances such as air pollution and low vegetation cover [71,90,91]. Omnivores are the second most rich-feeding guilds. Again, they have a wildly distributed guild because omnivores have an affinity to utilize natural resources and food [63]. Our study also supported that they extend themselves in urban areas with high numbers in both spatial and temporal gradients [82,92,93]. Granivorous and omnivorous tend to colonize the degraded agricultural landscape [94]. However, our study explained that only granivorous showed colonization in the mosaic landscape of cropland. In contrast, omnivores showed maximum colonization in water bodies, followed by residential and sal forest areas. The colonization of granivorous in cropland because the open habitats provide abundant seed grain [95,96]. Fruiting plants are regulated by frugivorous birds [63,97]. We observed that the abundance of frugivorous birds is higher in forest areas than in residential areas because of the high fruit plant diversity, such as Janum (*Syzygium cumini*), Kendu (*Diospyros melanoxylon*), etc., scattered in this landscape which attracted a large number of fruit-eating birds. Moreover, several studies indicated fruiting trees attract frugivorous birds in urban areas [98–101].

Nectarivores were preferably low in number compared to other guilds. They prefer open habitats and are regulated by flowering plants during the blooming season [102]. We witnessed that their numbers were comparatively higher in residential areas than in other areas because nectarivores were regulated by the blooming of banana (*Musa* sp.), papaya (*Carica papaya*), and *Hibiscus* sp. and *Lantana camara*.

## 5. Conclusions

We observed that the diversity in the habitat regulated the birds' diversity. Insectivorous birds were the highest feeding guilds, followed by omnivorous and granivorous. The cropland habitat regulated bird feeding guilds and diversity. Therefore, agricultural and degraded landscapes like cropland played an important role in maintaining bird diversity. Therefore, proper urban planning is required to protect bird diversity in the human-dominated landscape.

In eastern India, it is one of the first kinds of study which relied on identifying the bird compositions and diversity across the semi-urban landscape. Our study assessing the role of habitat that influences overall bird abundance, richness, and diversity would not capture differences in environmental factors (biotic and abiotic) requirements of different species/compositions across all habitats. Therefore, adding environmental factors into consideration may help to improve the findings. Further study is required to identify the major spatial and temporal drivers of bird diversity/composition and conservation problems in such developing cities.

**Author Contributions:** H.S.P. conceived the study. R.K., A.K. and A.G. collected the data. R.K. and H.S.P. performed the analyses. R.K. and H.S.P. wrote the first draft of the paper. A.K., A.G. and R.K.M. revised the manuscript. All authors have read and agreed to the published version of the manuscript.

**Funding:** This research received no external funding.

**Institutional Review Board Statement:** Not applicable.

**Data Availability Statement:** The data presented in this study are available on request from the corresponding author.

**Acknowledgments:** We are thankful to the Odisha Forest department and administrative authority of Maharaja Sriram Chandra Bhanjadeo University (formerly North Orissa University) for conducting the survey. We would like to thank Russell J. Grey for his technical advice on the use of R software.

**Conflicts of Interest:** The authors declare no conflict of interest.

## Appendix A

**Table A1.** Checklist and abundance of birds in the study area. NT = Near Threatened, LC = Least Concern, CL = Cropland, RA = Residential Areas, SF = Sal Forest, WB = Waterbodies, F = Frugivorous, G = Granivorous, I = Insectivorous, O = Omnivorous, C = Carnivorous.

| Family | Scientific Name | Common Name | Abbreviation of Common Name | IUCN Status | Guild | CL | RA | SF | WB |
|---|---|---|---|---|---|---|---|---|---|
| Psittacidae | *Psittacula eupatria* | Alexandrine Parakeet | ap | NT | F | 0 | 32 | 15 | 0 |
| Alaudidae | *Eremopterix griseus* | Ashy-crowned Sparrow Lark | acsl | LC | G | 17 | 0 | 0 | 0 |
| Cisticolidae | *Prinia socialis* | Ashy Prinia | asp | LC | I | 17 | 0 | 0 | 0 |
| Cuculidae | *Eudynamyscolopaceus* | Asian Koel | ak | LC | O | 0 | 11 | 6 | 0 |
| Ciconiidae | *Anastomus oscitans* | Asian Openbill | aop | LC | C | 24 | 0 | 0 | 101 |
| Apodidae | *Cypsiurus balasiensis* | Asian Palm Swift | aps | LC | I | 74 | 28 | 0 | 64 |
| Monarchidae | *Terpsiphone paradisi* | Indian Paradise-flycatcher | ipfc | LC | I | 0 | 0 | 8 | 0 |
| Sturnidae | *Gracupica contra* | Asian Pied Starling | apst | LC | O | 103 | 82 | 27 | 59 |
| Sturnidae | *Acridotheres ginginianus* | Bank Myna | bm | LC | O | 0 | 53 | 0 | 0 |
| Hirundinidae | *Hirundo rustica* | Barn Swallow | bs | LC | I | 43 | 0 | 0 | 28 |
| Ploceidae | *Plocus philippinus* | Baya Weaver | bw | LC | G | 41 | 0 | 0 | 0 |
| Laniidae. | *Lanius vittatus* | Bay-backed Shrike | bbs | LC | C | 0 | 0 | 9 | 0 |
| Dirucadae | *Dicrurus macrocercus* | Black Drongo | bd | LC | I | 40 | 37 | 38 | 14 |
| Estrildidae | *Lonchura malacca* | Black-headed Munia | bhm | LC | G | 36 | 0 | 0 | 0 |
| Accipitridae | *Milvus migrans* | Black kite | bk | LC | C | 4 | 15 | 0 | 4 |
| Accipitridae | *Elanus axillaris* | Black-shouldered Kite | bsk | LC | C | 6 | 8 | 0 | 0 |
| Campephagidae | *Lalage melanoptera* | Black-headed Cuckooshrike | bhcs | LC | O | 0 | 0 | 4 | 0 |
| Oriolidae | *Oriolus xanthornus* | Black-hooded Oriole | bho | LC | O | 0 | 23 | 16 | 11 |
| Oriolidae | *Oriolus chinensis* | Black-naped Oriole | bno | LC | O | 0 | 0 | 10 | 0 |
| Picidae | *Dinopium benghalense* | Black-rumped Woodpecker | brw | LC | I | 0 | 0 | 12 | 0 |
| Columbidae | *Columba livia* | Rock Dove | rd | LC | G | 90 | 191 | 0 | 0 |
| Megalaimidae | *Psilopogon asiatica* | Blue-throated Barbet | btb | LC | F | 0 | 0 | 13 | 0 |
| Meropidae | *Nyctyornis athertoni* | Blue-bearded Bee-eater | bbbe | LC | I | 0 | 0 | 3 | 0 |
| Turnicidae | *Turnix sylvaticus* | Common Buttonquail | cbq | LC | O | 26 | 0 | 0 | 0 |
| Sturnidae | *Sturnia pagodarum* | Brahminy Starling | bst | LC | O | 12 | 34 | 37 | 0 |
| Dirucadae | *Dicrurus aeneus* | Bronzed Drongo | brd | LC | I | 3 | 13 | 10 | 0 |
| Jacanidae | *Metopidius indicus* | Bronze-winged Jacana | bwj | LC | I | 0 | 0 | 0 | 81 |
| Laniidae. | *Lanius cristatus* | Brown Shrike | brs | LC | I | 1 | 13 | 0 | 0 |
| Megalaimidae | *Psilopogon zeylanicus* | Brown-headed Barbet | bhb | LC | F | 0 | 0 | 10 | 0 |
| Ardeidae | *Bubulcus ibis* | Cattle Egret | ce | LC | I | 147 | 118 | 96 | 0 |
| Meropidae | *Merops leschenaulti* | Chestnut-headed Bee-eater | chbe | LC | I | 45 | 0 | 24 | 0 |
| Sturnidae | *Sturnia malabarica* | Chestnut-tailed Starling | cts | LC | O | 20 | 117 | 121 | 0 |
| Phylloscopidae | *Phylloscopus collybita* | Common Chiffchaff | cc | LC | I | 3 | 7 | 0 | 0 |
| Rallidae | *Fulica atra* | Common Coot | cct | LC | O | 0 | 0 | 0 | 83 |
| Cuculidae | *Hierococcyx varius* | Common Hawk-Cuckoo | chc | LC | I | 0 | 2 | 3 | 0 |
| Upupidae | *Upupa epops* | Eurasian Hoopoe | eh | LC | I | 24 | 0 | 0 | 0 |
| Aegithinidae | *Aegithina tiphia* | Common Iora | ci | LC | I | 0 | 0 | 13 | 0 |
| Alcedinidae | *Alcedo atthis* | Common Kingfisher | ck | LC | C | 9 | 8 | 0 | 40 |
| Rallidae | *Gallinula chloropus* | Common Moorhen | cmh | LC | O | 0 | 8 | 0 | 58 |
| Sturnidae | *Acridotheres tristis* | Common Myna | cm | LC | O | 76 | 66 | 64 | 57 |
| Scolopacidae | *Actitis hypoleucos* | Common Sandpiper | csp | LC | I | 17 | 0 | 0 | 0 |
| Scolopacidae | *Gallinago gallinago* | Common Snipe | csn | LC | I | 12 | 0 | 0 | 13 |
| Cisticolidae | *Orthotomus sutorius* | Common Tailorbird | ctb | LC | I | 5 | 35 | 0 | 0 |
| Megalaimidae | *Megalaima haemacephala* | Coppersmith Barbet | csb | LC | F | 0 | 18 | 25 | 0 |
| Anatidae | *Nettapus coromandelianus* | Indian Pygmy Goose | ipg | LC | O | 0 | 0 | 0 | 166 |
| Accipitridae | *Spilornis cheela* | Crested Serpent-Eagle | cse | LC | C | 0 | 0 | 5 | 0 |
| Motacillidae | *Motacilla citreola* | Citrine Wagtail | cw | LC | I | 13 | 9 | 7 | 2 |
| Phylloscopidae | *Phylloscopus fuscatus* | Dusky Warbler | dw | LC | I | 0 | 11 | 2 | 0 |
| Columbidae | *Streptopelia tranquebarica* | Red Collared-Dove | rcd | LC | G | 10 | 0 | 0 | 0 |
| Columbidae | *Streptopelia decaocto* | Eurasian Collared-Dove | ecd | LC | G | 52 | 0 | 54 | 0 |
| Oriolidae | *Oriolus oriolus* | Eurasian Golden Oriole | ego | LC | O | 0 | 23 | 18 | 6 |
| Motacillidae | *Dendronanthus indicus* | Forest Wagtail | fw | LC | I | 0 | 0 | 7 | 0 |
| Picidae | *Dendrocopos macei* | Fulvous-breasted Woodpecker | fbw | LC | I | 0 | 0 | 10 | 0 |
| Cuculidae | *Centropus sinensis* | Greater Coucal | gc | LC | O | 7 | 5 | 7 | 0 |
| Picidae | *Chrysocolaptes guttcristatus* | Greater Golden-backed Woodpecker | ggbw | LC | I | 0 | 0 | 12 | 0 |
| Meropidae | *Merops orientalis* | Green Bee-eater | gbe | LC | I | 51 | 81 | 66 | 30 |
| Cuculidae | *Phaenicophaeus tristis* | Green-billed Malkoha | gbm | LC | C | 0 | 0 | 2 | 0 |
| Motacillidae | *Motacilla cinerea* | Grey Wagtail | gw | LC | I | 10 | 10 | 7 | 0 |
| Corvidae | *Corvus splendens* | House Crow | hc | LC | O | 0 | 6 | 0 | 47 |
| Passeridae | *Passer domesticus* | House Sparrow | hs | LC | G | 37 | 79 | 0 | 0 |
| Apodidae | *Apus nipalensis* | House Swift | hsw | LC | I | 0 | 58 | 0 | 0 |
| Caprimulg | *Caprimulgus asiaticus* | Indian Nightjar | inj | LC | I | 8 | 0 | 0 | 0 |

**Table A1.** *Cont.*

| Family | Scientific Name | Common Name | Abbreviation of Common Name | IUCN Status | Guild | CL | RA | SF | WB |
|---|---|---|---|---|---|---|---|---|---|
| Ardeidae | *Ardeola grayii* | Indian Pond-Heron | ph | LC | C | 21 | 21 | 0 | 29 |
| Muscicapidae | *Saxicoloides fulicatus* | Indian Robin | iro | LC | I | 0 | 0 | 21 | 0 |
| Coraciidae | *Coracias benghalensis* | Indian Roller | irl | LC | C | 8 | 0 | 0 | 0 |
| Cuculidae | *Clamator jacobinus* | Jacobin Cuckoo | jc | LC | I | 4 | 7 | 0 | 0 |
| Accipitridae | *Aviceda jerdoni* | Jerdon's Baza | jb | LC | C | 0 | 0 | 4 | 0 |
| Timaliidae | *Turoides striata* | Jungle Babbler | jbb | LC | O | 27 | 50 | 0 | 0 |
| Sturnidae | *Acridotheres fuscus* | Jungle Myna | jm | LC | O | 29 | 57 | 0 | 0 |
| Campephagidae | *Coracina javensis* | Large Cuckooshrike | lcs | LC | I | 7 | 9 | 0 | 0 |
| Ardeidae | *Ardea alba* | Great Egret | ge | LC | C | 0 | 0 | 0 | 14 |
| Anatidae | *Dendrocygna javanica* | Lesser Whistling Duck | lwd | LC | O | 0 | 0 | 0 | 461 |
| Ardeidae | *Egretta garzetta* | Little Egret | leg | LC | C | 0 | 0 | 0 | 15 |
| Podicipedidae | *Tachybaptus ruficollis* | Little Grebe | lg | LC | C | 0 | 7 | 0 | 275 |
| Muscicapidae | *Copsychus saularis* | Oriental Magpie-Robin | omr | LC | I | 0 | 26 | 0 | 0 |
| Alaudidae | *Alauda gulgula* | Oriental Sky Lark | osl | LC | G | 26 | 0 | 0 | 0 |
| Zosteropidae | *Zosterops palpebrosus* | Oriental White-eye | owe | LC | O | 0 | 8 | 44 | 0 |
| Motacillidae | *Anthus rufulus* | Paddyfield Pipit | pfp | LC | I | 32 | 0 | 0 | 0 |
| Jacanidae | *Hydrophasianus chirurgus* | Pheasant-tailed Jacana | ptj | LC | I | 0 | 0 | 0 | 61 |
| Alcedinidae | *Ceryle rudis* | Pied Kingfisher | pk | LC | C | 0 | 0 | 0 | 4 |
| Cisticolidae | *Prinia inornata* | Plain Prinia | pp | LC | I | 21 | 9 | 0 | 0 |
| Psittacidae | *Psittacula cyanocephala* | Pulm-headed Parakeet | php | LC | F | 0 | 0 | 22 | 0 |
| Nectariniidae | *Cinnyris asiaticus* | Purple Sunbird | psb | LC | N | 38 | 60 | 0 | 15 |
| Nectariniidae | *Leptocoma zeylonica* | Purple-rumped Sunbird | prsb | LC | N | 28 | 57 | 0 | 15 |
| Dirucadae | *Dicrurus paradiseus* | Greater Racket-tailed Drongo | lrtd | LC | I | 0 | 0 | 14 | 0 |
| Hirundininae | *Cecropis daurica* | Red-rumped Swallow | rrsw | LC | I | 40 | 0 | 0 | 0 |
| Pycnonotidae | *Pycnonotus cafer* | Red-vented Bulbul | rvb | LC | O | 20 | 45 | 98 | 20 |
| Pycnonotidae | *Pycnonotus jocosus* | Red-whiskered Bulbul | rwb | LC | O | 11 | 18 | 52 | 7 |
| Psittacidae | *Psittacula krameri* | Rose-ringed Parakeet | rrp | LC | F | 18 | 24 | 75 | 0 |
| Sturnidae | *Pastor roseus* | Rosy Starling | rs | LC | O | 36 | 63 | 0 | 0 |
| Corvidae | *Dendrocitta vagabunda* | Rufous Treepie | rtp | LC | O | 0 | 37 | 32 | 10 |
| Picidae | *Micropternus brachyurus* | Rufous Woodpecker | rw | LC | I | 0 | 0 | 8 | 0 |
| Estrildidae | *Lonchura punctulata* | Scaly-breasted Munia | sbm | LC | G | 47 | 19 | 0 | 0 |
| Accipitridae | *Accipiter badius* | Shikra | sk | LC | C | 6 | 18 | 12 | 6 |
| Accipitridae | *Circaetus gallicus* | Short-toed Snake Eagle | stse | LC | C | 0 | 0 | 4 | 0 |
| Columbidae | *Spilopelia chinensis* | Spotted Dove | sd | LC | G | 68 | 51 | 54 | 19 |
| Strigidae | *Athene brama* | Spotted Owlet | so | LC | C | 0 | 17 | 6 | 3 |
| Alcedinidae | *Pelargopsis capensis* | Stork-billed Kingfisher | sbk | LC | C | 0 | 0 | 0 | 8 |
| Dicaeidae | *Dicaeum agile* | Thick-billed Flowerpecker | tbfp | LC | O | 0 | 0 | 32 | 0 |
| Muscicapidae | *Ficedula albicilla* | Taiga Flycatcher | tfc | LC | I | 4 | 0 | 0 | 0 |
| Muscicapidae | *Eumyias thalassinus* | Asian Verditer Flycatcher | avfc | LC | I | 4 | 0 | 8 | 0 |
| Rallidae | *Amaurornis phoenicurus* | White-breasted Waterhen | wbwh | LC | O | 25 | 26 | 0 | 0 |
| Alcedinidae | *Halcyon smyrnensis* | White-throated Kingfisher | wtk | LC | C | 12 | 20 | 0 | 16 |
| Motacillidae | *Motacilla alba* | White Wagtail | ww | LC | I | 15 | 12 | 0 | 0 |
| Dirucadae | *Dicrurus caerulescens* | White-bellied Drongo | wbd | LC | I | 0 | 0 | 7 | 0 |
| Artamidae | *Artamus cyanopterus* | Dusky Woodswallow | dws | LC | I | 0 | 41 | 0 | 0 |
| Sylviidae | *Chysomma sinense* | Yellow-eyed Babbler | yeb | LC | I | 11 | 9 | 0 | 0 |
| Columbidae | *Treron phoenicopterus* | Yellow-legged Green Pigeon | ylgp | LC | F | 33 | 0 | 0 | 0 |
| Cisticolidae | *Cisticola juncidis* | Zitting Cisticola | zsc | LC | I | 5 | 0 | 0 | 0 |
| Charadriidae | *Vanellus indicus* | Red-wattled lapwing | rwl | LC | I | 23 | 0 | 0 | 0 |
| Phalacrocoracidae | *Microcarbo niger* | Little Cormorant | lc | LC | C | 0 | 0 | 0 | 136 |
| Ardeidae | *Ardea purpurea* | Purple Heron | phr | LC | C | 0 | 0 | 0 | 14 |
| Chloropseidae | *Chloropsis aurifrons* | Golden-fonted Leafbird | gflb | LC | I | 0 | 0 | 14 | 0 |
| Campephagidae | *Pericrocotus flammeus* | Scarlet Minivet | sm | LC | I | 0 | 0 | 23 | 0 |
| Phasianidae | *Francolinus pondicerianus* | Grey Francolin | gf | LC | O | 26 | 0 | 0 | 0 |
| Pittidae | *Pitta brachyura* | Indian Pitta | ip | LC | I | 0 | 0 | 8 | 0 |
| Threskiornithidae | *Plegadis falcinellus* | Glossy Ibis | gi | LC | C | 32 | 0 | 0 | 0 |

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
