# Peer review of "Bird Assemblages in a Peri-Urban Landscape in Eastern India"

_2673-6004, doi:10.3390/birds3040026_

Round 1

Author Response

Reviewer 1:

Before responding to your comments, we would like to take this opportunity to thank you for your extensive review. All revisions recommended by you are in Track change mode in the revised manuscript.

General comments:

This study examines avian species richness and diversity in an area undergoing rapid urban expansion due to its economic emergence as an educational hub and which is also known for its forestry industry that has resulted in vast areas of deforestation. It is therefore timely that this article is written as it will add to the gradually increasing store of knowledge regarding the effect of urbanisation on avian communities in tropical and sub-tropical areas, a field of ecology that has long lagged behind its temperate zone counterparts.

The study itself appears well-planned and executed and the use of Bray-Curtis and Jaccard indices apt in comparing species between habitats. However, I would have liked more clarity with regard to the cropland/wasteland habitat; these would seem to be two very different types of habitat so and explanation as why they were combined would be useful. As Baripada is described as a rapidly growing town, it would also have been of interest to know whether any invasive species were recorded taking advantage of the changing landscapes.

Specific comments

Abstract:

In the main, the abstract is clear and informative as to the study following.

Page 1, line 12: Remove ‘about’ before 8 transects.

Ans:  Removed ‘about’ from the sentence.

Page 1, line 14: define NMDS.

Ans: Firstly, we started the sentence with `non-parametric multidimensional scaling [NMDS], and then a short description of ‘NMDS’ is added to the sentence.

Page 1, lines 20 & 21: I feel this could be re-worded to make it more definitive and less ambiguous.

Ans: As per suggestion we have reconstructed the sentences.

Introduction:

The introduction places the study in context and clearly states the aim of the project; however, I would like to see some clarification as to why the particular habitats were chosen, a definition of ‘wasteland’ and why this was combined with ‘cropland’.

Ans: Wastelands are the small habitat patches of unfertile cropland that remained uncultivated for longtime where wild herb or shrub grew over time. It’s difficult to differentiate from cropland. As per the other reviewer’s recommendation, we merged it into cropland habitat in the revised manuscript.

Page 1, lines 32 & 33: This sentence is unclear and needs to be re-worded.

Ans: We have clarified this line with proper references.

Page 2, lines 59-61: Please define and explain how and why these particular habitats were chosen.

Ans: We defined each habitat with a proper justification in our selection criteria as per suggestion. 

Methods & Materials:

Overall the Methods section flows and reads well with some clarity issues. Figure 1 and Table 1 are particularly useful.

Page 2, line 68: A temperature range for the specific seasons would be of interest; is 30°C the average annual temperature?

Ans: Summer is hot and humid (45°) , winter is dry and cold(10 C°) and annual temperature is 30° C.

Page 3, lines 96 & 97: This needs to be clearer; what was the minimum – one or two counts?

Ans: Now, we have clearly mentioned with a proper transect count at each place for each month.

Page 3, lines 99 & 100: Were birds identified by sight alone?

Ans: Yes. All the birds encountered in the field are identified visually with the help of binoculars by querying the field guidebook.  However, we used to take photographs of birds on the spot if we failed to identify any birds during the survey.  Later on, we took experts' help to identify the birds.

Results:

The results logically follow the steps presented in the methods section. Figures are clear and informative though I do question the necessity of Figure 4 (the statement of the analysis results suffice).

Page 5, lines 157 & 158: It would be interesting to know what the remaining three species were (117 sp. in total, 103 residents, 11 winter migrants = 114 sp.).

Ans: Thank you very much for this attention. The remaining three birds are summer visitors. We have clarified this clearly in the revised manuscript.

Discussion:

I feel the discussion section could benefit from a comparison of similar studies both in the temperate zones and, if available, in the tropical/sub-tropical areas of the world. Were the results as the authors expected, or have there been any surprises? Some suggestions as to how conservation and urban planning might incorporate results from studies such as these would also be beneficial.

Page 9 & 10, lines 244-247: This seems vague and part of it unnecessary. It needs to be made clear that the avian richness and diversity was highest in cropland/wasteland for this area specifically. I also don’t think the statement about certain bird species preferring cropland habitats occasionally is required; the case could be made that any number of bird species visits a particular habitat occasionally. The following sentence from lines 247-249 is more pertinent.

Ans: We removed ‘occasionally’ word from the revised manuscript.

Page 10, lines 251 & 252: This sentence needs to be clearer.

Ans: We have clarified this sentence clearly in the revised manuscript.

Page 10, lines 259 & 260: As above, therichness of bird richness?

Ans: Richness of birds. We have corrected the mistake.

Page 10, lines 273-275: Do you have species examples for this? Are these invasive species or local opportunistic species?

Ans: We have given species examples in the revised manuscript. All these are local opportunistic species.

Page 10, lines 276-279: Can you compare this result to other studies conducted in tropical or sub-tropical areas or is this in comparison to temperate zone studies?

Ans: We have incorporated your suggestions in the revised manuscript (Last paragraph, Discussion section).

Reviewer 2 Report

It is a pleasure for me to review the manuscript number “birds-1954080 ” titled “Bird assemblages in a peri-urban landscape in eastern India ” for  “Birds’ journal.

The paper is complete and very well written. However, there are suggestions that I must make concerning this work in order to improve the overall level of this manuscript to merit publication in this journal. 

1) Given the presence of some English mistakes (grammatical, typos, and run-on sentence). I suggest to check carefully the manuscript another time and make a revision by a native English or a specialist.

2)Line 8, I think you need a more general statement before writing about the tropic maybe some thing more general about urbanisation and biodiversity.

3)Line 14, Please add the full term of NMDS.

4) In the same line, that instead of the.

5) Lines 25-33, you introduction is great, however, as I said previously in your abstract, I think you don't have to limit yourself to tropics because it's a global problem, you can write in general about this phenomenon and specifying that it's especially the case in tropics.

6) Again, the first part of the introduction is very important and there is a lack of references, there are some recent interesting works dealing with the same theme, please add these relevant citations in this part or any other location in your manuscript you judge relevant :

*https://doi.org/10.1038/s41598-022-08654-7

*https://doi.org/10.1186/s40657-021-00280-7

*https://doi.org/10.25225/jvb.22027

*https://doi.org/10.2989/00306525.2022.2068691

*https://doi.org/10.3390/land10040434

*https://doi.org/10.1016/j.gecco.2022.e02217

*https://doi.org/10.1146/annurev-environ-112420-014642

*https://doi.org/10.1016/j.crvi.2017.07.002

*https://doi.org/10.3390/d14040253

*https://doi.org/10.3390/birds3010007

7) You can specify in the penultimate paragraph that there are also phenomena of adaptation of birds to different types of environments going as far as a modification of the phenotype (sometimes of the genotype), please refer to Belabed et al. (2014). The effect of urbanization on the phenotype of the Collared Dove (Streptopelia decaocto) in northeastern Algeria. Bull. Inst. Sci. Rabat. Sect. Sci. Vie 2014, 35, 155–164. And cite it.

8) In the last Paragraph of the introduction, it's better to clearly indicate your objective (eg. this study aimed at, the aim of the study...etc).

9)line 64, the 1st sentence doesn't read well, please rewrite it, for example this study was carried out in Baripada or was conducted in Baripada...

10) Lines 114-127, please write equations in an independant line to avoid overloading and make it more pleasant to read.

11) Line 144, please specify the citations according to the journal instructions.

12) Please cite R software and the package you used.

13) In my humble opinion, the way of showing results can be improved, I congratulate you for the tests used (which are quite adequate). However, the presentation of the rank-abundance, the curve is the same even if the number of birds is different, which is confusing, perhaps it should have been adapted to the number, what do you think about it ?

14) The part concerning Feeding guilds and functional diversity is a good idea, congratulations.

15) Line 288-289 same as my comment number 11

16) In the end of the discussion part, maybe it could be interesting to add at least 1 or 2 sentences to write about the limitations of your study and even your perspectives to improve it in the future.

17) I also thank you for your supplementary file, I know how much it was a hard work to do.

I wish the authors good luck for the publication process and for their future works.

Author Response

Reviewer 2:

We would like to thank you for your extensive reviews of the manuscript. Given below are our responses to the line-wise comments made by you. All revisions recommended by you are in Track change mode in the revised manuscript.

It is a pleasure for me to review the manuscript number “birds-1954080 ” titled “Bird assemblages in a peri-urban landscape in eastern India ” for  “Birds’ journal.

The paper is complete and very well written. However, there are suggestions that I must make concerning this work in order to improve the overall level of this manuscript to merit publication in this journal.

1) Given the presence of some English mistakes (grammatical, typos, and run-on sentence). I suggest to check carefully the manuscript another time and make a revision by a native English or a specialist.

Ans: We have checked carefully and revised throughout the manuscript to improve the language and quality.

2)Line 8, I think you need a more general statement before writing about the tropic maybe some thing more general about urbanisation and biodiversity.

Ans: We have added a few words regarding urbanisation and biodiversity loss across the globe as per suggestion.

3)Line 14, Please add the full term of NMDS.

Ans: In this regard [‘NMDS’], Reviewer 1 also mentioned the same question. Now, we have followed your instruction and defined it clearly with little description in the revised manuscript.    

4) In the same line, that instead of the.

Ans: We have incorporated your suggestion in the revised manuscript.

5) Lines 25-33, you introduction is great, however, as I said previously in your abstract, I think you don't have to limit yourself to tropics because it's a global problem, you can write in general about this phenomenon and specifying that it's especially the case in tropics.

Ans: In the revised manuscript, we have added valuable information to the first paragraph of our introductory part. We cited several papers which dealt with the consequences of urbanisation on bird diversity. In addition, the referred literature is really helpful for our paper.

6) Again, the first part of the introduction is very important and there is a lack of references, there are some recent interesting works dealing with the same theme, please add these relevant citations in this part or any other location in your manuscript you judge relevant :

*https://doi.org/10.1038/s41598-022-08654-7

*https://doi.org/10.1186/s40657-021-00280-7

*https://doi.org/10.25225/jvb.22027

*https://doi.org/10.2989/00306525.2022.2068691

*https://doi.org/10.3390/land10040434

*https://doi.org/10.1016/j.gecco.2022.e02217

*https://doi.org/10.1146/annurev-environ-112420-014642

*https://doi.org/10.1016/j.crvi.2017.07.002

*https://doi.org/10.3390/d14040253

*https://doi.org/10.3390/birds3010007

Ans: We have incorporated your suggestion in the revised manuscript.

7) You can specify in the penultimate paragraph that there are also phenomena of adaptation of birds to different types of environments going as far as a modification of the phenotype (sometimes of the genotype), please refer to Belabed et al. (2014). The effect of urbanization on the phenotype of the Collared Dove (Streptopelia decaocto) in northeastern Algeria. Bull. Inst. Sci. Rabat. Sect. Sci. Vie 2014, 35, 155–164. And cite it.

Ans: The referred information is really informative and relevant to our topic.  We have incorporated your suggestion in the revised manuscript.

8) In the last Paragraph of the introduction, it's better to clearly indicate your objective (eg. this study aimed at, the aim of the study...etc).

Ans: We have incorporated your suggestion in the revised manuscript.

9)line 64, the 1st sentence doesn't read well, please rewrite it, for example this study was carried out in Baripada or was conducted in Baripada...

Page 2, Line 64: We have modified our sentence as per instruction in the revised manuscript. 

10) Lines 114-127, please write equations in an independant line to avoid overloading and make it more pleasant to read.

Ans: Earlier, the equations were written in a single sentence which was difficult to read. As per suggestion, we wrote each mathematical equation in an independent line to avoid overlapping and made it understandable for everyone in the revised manuscript.

11) Line 144, please specify the citations according to the journal instructions.

Ans: After revision, we have rectified our mistake and provided a proper citations format according to the journal instructions.  

12) Please cite R software and the package you used.

Ans: We missed to provide proper citation for R packages. Now we have provided a proper citation for R software and packages.  

R Core Team (2013). R: A language and environment for statistical analysis computing. R Foundation for Statistical Computing, Vienna, Australia. URL https://www.R-project.org/.

Kindt, R., & Coe, R. (2005). Tree diversity analysis: a manual and software for common statistical methods for ecological and biodiversity studies. World Agroforestry Centre (ICRAF), Nairobi. ISBN 92-9059-179-X.

Chen H (2022). VennDiagram: Generate High-Resolution Venn and Euler Plots_. R package version   1.7.3, <https://CRAN.R-project.org/package=VennDiagram>.

Barter R, Yu B (2017). _superheat: A Graphical Tool for Exploring Complex Datasets Using   Heatmaps_. R package version 0.1.0, <https://CRAN.R-project.org/package=superheat>.

13) In my humble opinion, the way of showing results can be improved, I congratulate you for the tests used (which are quite adequate). However, the presentation of the rank-abundance, the curve is the same even if the number of birds is different, which is confusing, perhaps it should have been adapted to the number, what do you think about it ?

Ans: As per your suggestion, we have changed the figure (Rank Abundance Plot) in the revised manuscript

14) The part concerning Feeding guilds and functional diversity is a good idea, congratulations.

Ans: Thank you very much for your compliments.

15) Line 288-289 same as my comment number 11

Ans: We incorporated your suggestion in the revised manuscript.

16) In the end of the discussion part, maybe it could be interesting to add at least 1 or 2 sentences to write about the limitations of your study and even your perspectives to improve it in the future.

Ans: Page 10, Point 16, Line 298: We added a few sentences about the limitations of our study and future research concepts in the revised manuscript (Last paragraph).

17) I also thank you for your supplementary file, I know how much it was a hard work to do.

I wish the authors good luck for the publication process and for their future works.

Thank you very much for your rigorous review and compliments.